# Selection of Higher Order Lamb Wave Mode for Assessment of Pipeline Corrosion

Donatas Cirtautas *, Vykintas Samaitis [ID], Liudas Mažeika, Renaldas Raišutis [ID] and Egidijus Žukauskas

Prof. K. Baršauskas Ultrasound Research Institute, Kaunas University of Technology, 44249 Kaunas, Lithuania; vykintas.samaitis@ktu.lt (V.S.); liudas.mazeika@ktu.lt (L.M.); renaldas.raisutis@ktu.lt (R.R.); e.zukauskas@ktu.lt (E.Ž.)
* Correspondence: donatas.cirtautas@ktu.edu

**Abstract:** Hidden corrosion defects can lead to dangerous accidents such as leakage of toxic materials causing extreme environmental and economic consequences. Ultrasonic guided waves showed good potential detecting distributed corrosion in pipeline networks at sufficiently large distances. To simplify signal analysis, traditional guided wave methods use low frequencies where only fundamental modes exist; hence, the small, localized defects are usually barely detectable. Novel techniques frequently use higher order guided wave modes that propagate around the circumference of the pipe and are more sensitive to the localized changes in the wall thickness. However current high order mode guided wave technology commonly uses either non-dispersive shear modes or higher order mode cluster (HOMC) waves that are mostly sensitive to surface defects. As the number of application cases of high order modes to corrosion detection is still limited, a huge potential is available to seek for other modes that can offer improved resolution and sensitivity to localized corrosion type defects. The objective of this work was to investigate higher order modes for corrosion detection and to determine the most promising ones in sense of excitability, leakage losses, propagation distance, and potential simplicity in the analysis. The selection of the proper mode is discussed with the support of phase and group velocity dispersion curves, out of plane and in plane distributions over the thickness and on surface of the sample, and leakage losses due to water load. The analysis led to selection of symmetric $S_3$ mode, while the excitation of it was demonstrated through finite element simulations, taking into account the size of phased array aperture and apodization law and considering two-sided mode generation. Finally, theoretical estimations were confirmed with experiments, demonstrating the ability to generate and receive selected mode. It was shown that $S_3$ mode is a good candidate for corrosion screening around the circumference of the pipe, as it has sufficient propagation distance, can be generated with conventional ultrasonic (UT) phased arrays, has sufficiently high group velocity to be distinguished from co-existing modes, and is sensitive to the loss of wall thickness.

**Keywords:** guided waves; higher-order mode; corrosion detection; symmetrical mode; mode selective excitation; particle displacement; SAFE; FEM; group velocity; phase velocity

## 1. Introduction

Corrosion is one of the most common degradation mechanisms of metallic engineering infrastructure. The pipelines of petroleum, chemical, and gas industries carry products that contain corrosive substances leading to progressive degradation of the equipment. EU reports at least 137 major accidents in petroleum refineries since 1984 while around 20% of them are corrosion related, leading to annual global losses up to USD 2.5 trillion [1,2]. To avoid severe environmental and economic consequences, the periodic inspection of corrosion rates is a must to ensure long-term integrity of the assets [3].

Ultrasonic testing stands out among the classic corrosion detection techniques, such as linear polarization resistance [4], electrochemical noise [5], or weight loss measurements [6],

as the direct and non-intrusive method that allows to assess the corrosion. Classic non-destructive testing (NDT) methods use conventional probes based on wall thickness measurements. Spot measurements have high sensitivity to defects but at the cost of inspection time, so it is almost impossible to inspect long pipelines due low inspection coverage. Such methods are quite well established and allow to track wall thickness changes of micron levels ($\pm 100$ μm/year) [7] with repeatability of up to 40 nm [8]. However, being essentially slow due to manual nature, such methods cannot cover large areas and may miss randomly distributed, concentrated corrosion [9–13]. In pipelines, mostly localized corrosion defects with small radius are accelerated by flow which assists penetration causing leakage. However, the locations of degradation usually cannot be predicted, and thus, large area screening techniques are preferable to achieve sufficient inspection coverage. Ultrasonic guided waves showed good potential detecting various defects in pipeline networks at sufficiently large distances. Conventional guided wave methods use low frequencies where only fundamental modes exist [14,15]. Low frequency guided waves have less attenuation and allow to inspect buried or covered pipe sections at distances up to 100 m. While low frequency guided waves are less sensitive to small defects, the reflectors must be sharp and large enough in order to detect them. In case of small localized defects or smooth wall thinning, the responses produced by the flaws are usually barely detectable by currently existing guided wave technology, which could essentially benefit from detecting defects producing weak responses [16,17].

Novel techniques based on thickness loss measurement around the circumference of the pipe emerged as a tool which allows more accurate yet still relatively large scale assessment of the corrosion [18,19]. Such techniques frequently use high order guided wave modes that propagate around the circumference of the pipe and are sensitive to the changes in wall thickness [20,21]. In such way, the entire circumference of the pipe can be inspected from a single transducer position.

For example, in a recent investigation P. Khalili analysed different corrosion inspection techniques including Multi-skip, Creeping Head-Wave, Higher Order Mode Cluster, symmetric $S_0$ at 15 MHz·mm, and shear-horizontal $SH_0$ and $SH_1$ at 3 MHz·mm waves. Different wave characteristics were investigated such as attenuation in liquid loading and coating, surface roughness and transmission reflection coefficients in defective zone, or interaction with T-joints. Each technique was able to achieve some satisfying results, however, with some limitations. For example, Multi-Skip and Creeping Head Waves are found to be sensitive to the surface condition, and most importantly, they are affected by liquid loading. Higher Order Mode Cluster with targeted $A_1$ wave mode has reduced sensitivity to surface defects and is mostly used for inspection of pipe segments under support or at T-joints. With other techniques authors exploited $SH_1$ waves; however, parasitic $SH_0$ was excited as well which in some cases may complicate the analysis. Despite presence of parasitic modes, authors showed that the $SH_1$ mode was sensitive to gradual and smaller defects down to 10% of wall thickness loss. In a follow-up paper, P. Khalili, and F. Cegla continued research focused on the $SH_1$ wave for inspection in a 10 mm aluminum plate detecting crack-like defects. The major down side of the proposed set-up is the requirement of electromagnetic acoustic transducers (EMAT) in order to generate an SH wave [22–24]. Overall, from results obtained in both studies, authors suggested to use combination of two or more different methods for reliable corrosion detection [25,26].

In addition, similar conclusions were drawn in the following research, where P. Khalili used isolated $A_1$ wave mode at high frequency such as 2.25 MHz at 10 mm aluminum plate. Using an angled probe excitation, the authors achieved a pure asymmetric $A_1$ wave mode excitation with low dispersion. However, the results showed that the excited wave mode is not sensitive to some surface defects [27].

V. Serey et al. focused on single mode excitation inside aluminum bars and offered a new technique which uses a burst of signals with up to eight transducers in the cross section. The setup allows to excite selective asymmetric modes in aluminum bars using mode generation in the harmonic regime. The results showed that excited wave modes

at 30 KHz were capable of detecting simulated defects. The investigation analyzed only asymmetric fundamental wave modes [28]. S. Park used an approach comparable to V. Serey. Authors offered a similar technique of angled probe excitation with single and dual wedges for generation of asymmetrical higher order modes at 1 MHz. $A_0$ until $A_5$ modes were produced on a StainlessSteel304 plate [29].

Recent studies showed that each method has its limitations and encounters problems in mode generation and isolation. Some existing techniques mostly use PZT (Lead Zirconate Titanate) transducers for asymmetric wave mode excitation, while other techniques that include excitation of shear modes require EMATs. Hence, there is lack of investigations demonstrating the application of high order symmetrical modes for corrosion detection. Symmetrical modes usually are less sensitive to liquid load and hence can ensure larger inspection distances [30,31]. The advanced techniques with higher frequency and selective higher order symmetric mode generation could increase defect detectability and inspection coverage. However, in such case, all issues related to higher mode selection, wave excitability, detectability, sensitivity to defects, and leakage losses should be considered.

The objective of this work was to investigate higher order symmetric modes for detection of corrosion defects and to determine the ones which are most promising in sense of excitability, leakage losses, propagation distance, and potential simplicity in signal analysis. In this paper, the selection of proper symmetric mode and excitation frequency is discussed, taking into account phase and group velocity dispersion curves. Desired wave mode excitability and detectability are analyzed using out of plane and in plane distributions on the surface of the pipe wall. The expected leakage losses are investigated through the analysis of particle displacement distributions over thickness of pipe wall and assessed in case of water load. Finally, approaches to generating pure $S_3$ mode at 1 MHz are proposed and discussed, demonstrating how different parameters, such as size of aperture, apodization, and two-sided excitation affect the mode purity. Theoretical investigations based on semi-analytical finite element (SAFE) and finite element (FE) modelling are confirmed with experiments on an SteelAlloy plate. Investigation demonstrated that $S_3$ mode at 1 MHz can be generated without parasitic modes in the FE model, which will be exploited in further studies analyzing the interaction between $S_3$ mode and corrosion type defects. The experiments demonstrated that due to a limited array aperture, other modes such as $S_2$ and $S_4$ are simultaneously generated. Nevertheless, the selected $S_3$ mode exhibits high group velocity, so it is expected that co-existing modes will not overlap at sufficient propagation distances. Overall, it is demonstrated that $S_3$ mode can be used for medium range inspection at distances up to 2 m, which is typical of pipes with outer diameter up to 0.64 m.

## 2. Object under Investigation

In this paper, we focus on a 9 mm thickness SteelAlloy1020 plate that mimics unrolled pipe with an outer diameter of 636.6 mm and a circumference of 2 m. The results presented by other authors suggest that Rayleigh–Lame dispersion equations for plates can be used as approximation to obtain dispersion relations in the circumferential direction of the thin cylindrical shell [19,32,33]. This is valid for thin wall pipes, where the ratio between inner ($r$) and outer ($R$) radius is $r/R > 0.95$. Other papers demonstrated that dispersion curves of plates are similar to hollow cylinders for pipe thickness ($h$) of radius ($R$) ratios less than 20% [34]. In this paper, we consider cases where $r/R > 0.97$, while $h/R \sim 3\%$. Hence the large radius thin-walled pipe can be considered similar to a plate, while the effect of curvature can be most noticeable at low frequencies only. During all calculations, the material properties of SteelAlloy1020 were considered as follows: density 7850 kg/m$^3$, Young's modulus 207 GPa, and Poisson's ratio 0.3. Further details concerning the dimensions of the structure and arrangement of the probes are presented in the following chapters next to the description of the finite element models and experimental set-ups.

## 3. Selection of Appropriate Mode for Assessment of Pipe Corrosion

Selection of the appropriate mode for inspection is one of the key aspects in the success of corrosion level assessment. The mode selection criteria should consider following aspects: mode excitability and receptibility, leakage losses, attenuation, dispersion, and velocity. Each and every aspect of mode selection is discussed further in the subsequent sections.

### 3.1. Dispersion Curves of the Considered Structure

At first stage of the investigations, semi-analytical finite element technique (SAFE) was used to calculate existing dispersion curves, particle displacements, and leakage losses on the unrolled SteelAlloy1020 pipe [35]. The correctness of such approach was stated by other authors [33,34] as it was mentioned above; however, we carried out additional verification by modelling using the FE method, and the validity of plate approximation will be demonstrated later in this chapter, providing a comparison of simulation results between pipes and plates.

In this investigation, dispersion curves were calculated using a 1D plate model and dividing the cross section of the 9 mm plate into 20 discrete elements each with a size of 0.45 mm. Each element was composed of three nodes. Such fine mesh size was deliberately selected as sufficient for the analysis of the displacement profiles of the propagating modes which will be presented later in this chapter [36]. The material properties were used as described in previous chapter. The calculated dispersion characteristics are presented in Figure 1. The dispersion characteristics presented below are considered for the structures exposed to air. The propagation of the modes in pipes filled with water are in some sense different, as the interface between the pipe and water is no longer traction-free. In such case, due to energy leakage to the surrounding media, an interface quasi-Scholte wave starts to propagate in the solid–liquid boundary [37]. This mode has negligible attenuation, short wavelength, and potentially increased sensitivity to small defects. However, at sufficiently high frequencies, it is known to have dominant in-plane displacements in liquid media and thus can no longer be detected from surface of solid structure [38]. The dispersion curves presented in this paper does not consider quasi-Scholte waves as the main purpose of the paper is to explore high order symmetric modes at frequencies, where the quasi-Scholte waves are no longer detectable and possess late arrival time to the sensor.

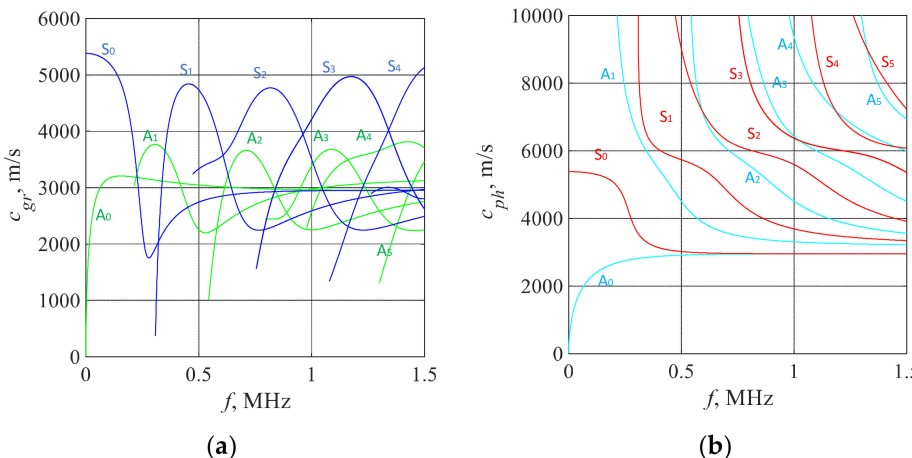

**Figure 1.** Dispersion curves in a 9 mm SteelAlloy1020 plate: (**a**) group velocity; (**b**) phase velocity.

From the results it can be observed that six asymmetrical and six symmetrical modes exist at frequencies up to 1.5 MHz. Symmetric modes have higher group velocities that peak at certain frequencies, for example, $S_1$ mode at 455 kHz, $S_2$—at 817 kHz and $S_3$—at 1.173 MHz (see Figure 1a). Each peak in group velocity of symmetrical modes corresponds to low-dispersion zone on phase velocity dispersion curves (see Figure 1b). It is preferable to use the mode with fastest group velocity (Figure 1a) as such mode will arrive to the

receiver first, simplifying the signal analysis. In this sense the most promising are frequency ranges where group velocities of symmetric modes peaks. However, it is known that at such frequencies the out of plane displacements of particular modes are usually poor, making them difficult to excite and receive using longitudinal ultrasonic transducers [39]. Moreover, in order to preserve the sensitivity to a change in pipe wall thickness that will result in measurable change in group velocity, the excitation frequency should be situated at sloped part of group velocity curve where excited modes are dispersive. As a result, the excitation should be considered at slightly other frequencies (above or below the peak frequency) avoiding that particular zero displacement zone, but still keeping at the dominant group velocity range.

It is necessary to take into account that only the modes that possess a sufficient out-of-plane component in a particular frequency range can be excited using conventional longitudinal wave ultrasonic transducers, which are sensitive to out-of-plane wave motion. On the other hand, the modes which possess strong out-of-plane component will be strongly attenuated due to leakage losses as most of the pipelines are filled with various substances [40]. Therefore, from that point of view, it is necessary to select an optimal mode and corresponding frequency ranges where out-of-plane displacement is sufficient for generation and reception, but on the other hand, it must be sufficiently low in order to have reasonable leakage losses and propagation distance that can be suppressed by conventional ultrasonic amplifiers.

Following the observations presented above, the $S_3$ mode has highest group velocity at frequency band between 0.9 MHz and 1.35 MHz. In such case, the $S_3$ mode is expected to arrive first compared to other modes. At frequencies below 1 MHz, $S_3$ mode has quite dispersive behavior, and hence, it is expected to have high sensitivity to the change in wall thickness of the pipe at a cost of resolution to small defects.

Further parameters that can limit inspection using $S_3$ mode are the leakage losses, attenuation, excitability, and receptibility, which are further explored in subsequent section. All these parameters are related to distribution of particle displacements on the surface of the plate. It the next chapter, the advantages of using $S_3$ mode over neighboring modes such as $S_1$ and $S_2$ will be shown.

*3.2. Leakage Losses and Mode Displacements*

Particle velocity and displacement distribution on the surface of the pipe determine the excitability and detectability of the mode and its energy losses to the surrounding media. Conventional longitudinal wave transducers or phased arrays vibrate at a resonant frequency that corresponds to the motion in thickness direction of piezo ceramics. The in-plane displacements cannot be detected by such transducers if these are attached through coupling liquid perpendicularly to the direction of in plane motion. As a result, such probes can excite and receive modes that have a sufficient out-of-plane component. Symmetric modes usually possess quite minor out-of-plane displacements, and thus, only those symmetric modes that have sufficient out-of-plane displacements can be excited with conventional longitudinal wave probes. On the other hand, such probes are effective, generating asymmetric modes and possessing significant out-of-plane displacements. Hence, it can be expected that while trying to excite symmetric modes using the conventional longitudinal wave transducers, undesired asymmetrical modes will be generated also within the selected frequency band. The out-of-plane and in-plane distributions of mode amplitudes at the surface of the steel alloy were calculated using the SAFE method and the same material properties as described in the previous section. The results are presented in Figure 2.

The out of plane component calculations on the surface of steel alloy plate suggest that the $S_3$ mode has quite sufficient out of plane components in the frequency band around 1 MHz to be generated and detected with conventional phased arrays. It can be observed that at these frequencies, other modes such as $A_1$, $S_1$, or $A_2$ will be generated more effectively. From the out-of-plane displacement curves, it can be observed that the $S_2$

mode can be excited starting at around 1 MHz. At this frequency, it has similar out-of-plane components to the $S_3$ mode; however, the displacements rise significantly for $S_2$ mode above 1 MHz.

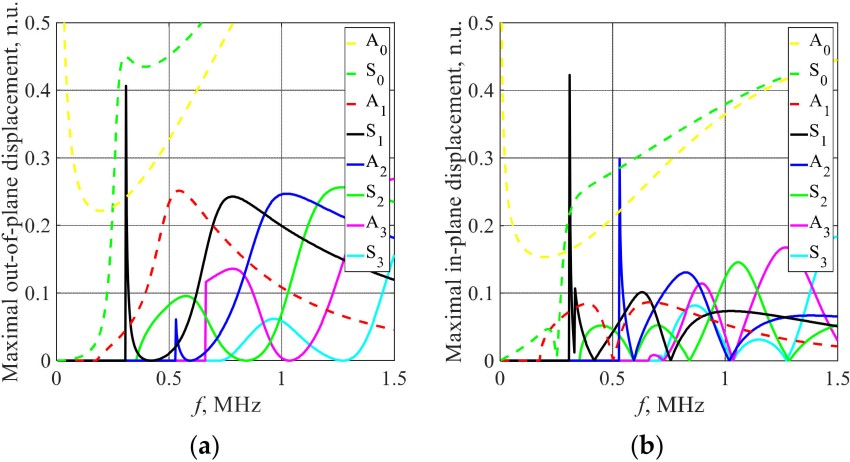

**Figure 2.** Maximal displacement versus frequency: (**a**) out-of-plane; (**b**) in-plane. The indices next to the mode shapes in the legend denote mode order.

The out-of-plane component of each specific mode is correlated with the leakage losses. The comparison between the amplitude distribution on the surface with leakage losses and the water loading is presented in Figure 3. As it can be observed, in Figure 3a, the displacement of normal components is strongly correlated with leakage losses, as expected, and less correlated to in-plane displacements as seen in Figure 3b. The leakage losses of the $S_3$ mode at 1 MHz is approximately 34 dB/m, and hence, the $S_3$ mode can be expected to be used at distances of up to 1.5–2 m, which is sufficient for screening of pipes at their cross section with outer diameters up to 640 mm. At such propagation distance, the total losses due to leakage are expected to be in the range 51 dB to 68 dB, which can be compensated by conventional ultrasonic amplifiers. It can be noted that other co-existing modes such as $A_1$, $S_1$, or $A_2$ have either similar (in case of $A_1$ mode) or significantly higher losses (in case of $S_1$ and $A_2$). The losses of $S_2$ mode are similar to $S_3$ at 1 MHz but are much more significant above this frequency.

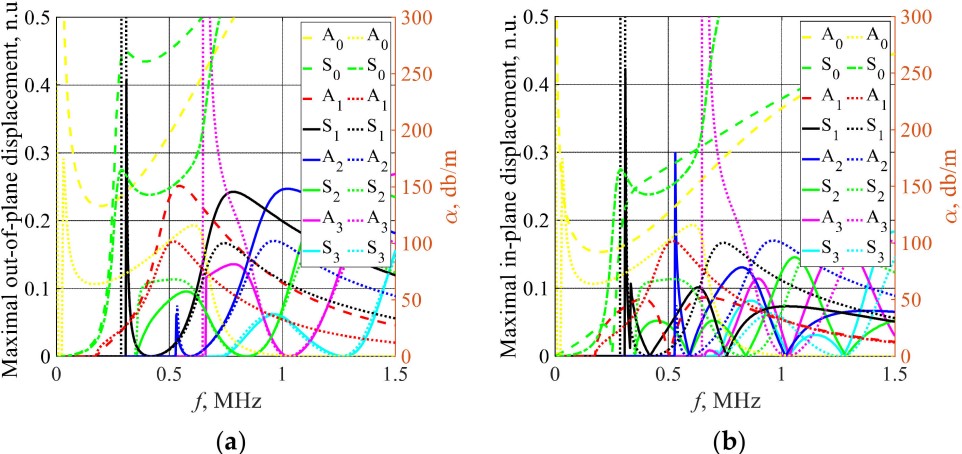

**Figure 3.** Comparison of displacement on surface (dashed and solid lines) and leakage losses (dash-dot and dotted lines): (**a**) out-of-plane component displacement and leakage losses versus frequency; (**b**) in-plane component amplitude and leakage losses vs. frequency. The indices next to the mode shapes in the legend denote mode order.

While the surface displacements show the excitability and leakage of the guided wave modes, the distribution over the thickness of plate determines which defects can be detected. To explore this in more detail, the displacement distribution in the cross section of the plate has been calculated for $S_3$ mode. The results are presented in Figure 4. It can be observed that out-of-plane displacements of $S_3$ mode exist at around 1 MHz frequency only. The out-of-plane displacements are zero at certain depths that correspond to middle thickness and depths of 1.5 mm and 7.5 mm according to the top surface.

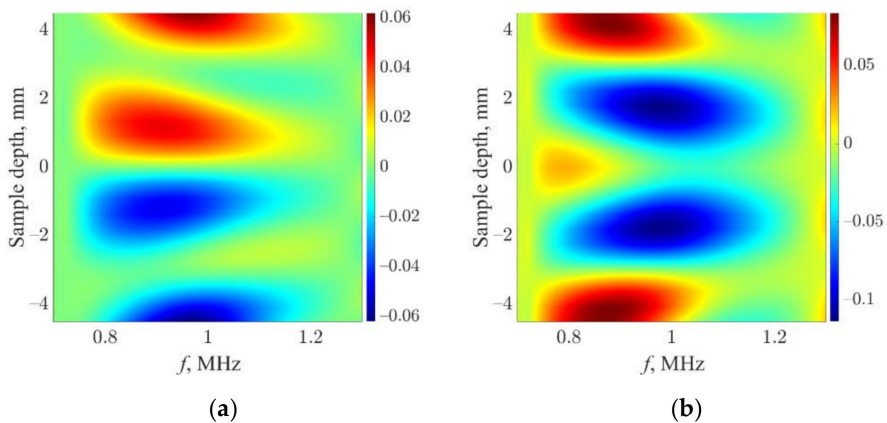

(**a**)             (**b**)

**Figure 4.** Particle displacement of $S_3$ mode across the thickness of steel plate versus frequency: (**a**) out-of-plane component; (**b**) in-plane component.

To explore the difference between the displacement distribution of the $S_3$ mode and other symmetrical and asymmetrical modes, the normalized displacement distributions of $A_0$-$A_3$ and $S_0$-$S_3$ modes were estimated at 1 MHz and presented in Figure 5. It can be observed in thickness cross-section of displacements that the particle displacement distribution of the $S_3$ mode varies inside the plate but also has sufficient particle displacement on the plate surface. The distribution of the waves across the thickness of the plate is not uniform and should be taken into account when the sensitivity to the different types of defects is analyzed. The results suggest that out-of-plane displacements of the $S_3$ mode peak at the surface of the plate and at depths corresponding to 1 mm and $-1$ mm with respect to middle thickness of the plate. All symmetrical modes in contrast to asymmetrical ones possess out of plane displacements close to zero at middle thickness. The $S_2$ mode has a strong in-plane component on the surface of the plate and relatively small leakage losses making it suitable for shallow defect detection. On the other hand, the group velocity of $S_2$ is quite low at 1 MHz which will complicate the analysis of arriving signals, while above 1 MHz, the leakage losses of this mode become unacceptable. In contrast, asymmetrical modes $A_1$ and $A_2$ have quite strong middle thickness out-of-plane displacements, but displacements at surface of the sample are quite low for $A_1$ and $A_3$ modes. The $A_2$ mode could be considered as good candidate for corrosion detection; however, essential attenuation limits its applicability.

The results presented above suggest that $S_3$ mode at 1 MHz could be used for corrosion assessment as it has high group velocity, out-of-plane displacements, and acceptable leakage losses for inspections up to 2 m. In subsequent sections, the excitation of the $S_3$ mode will be further explored using finite element model and experiments.

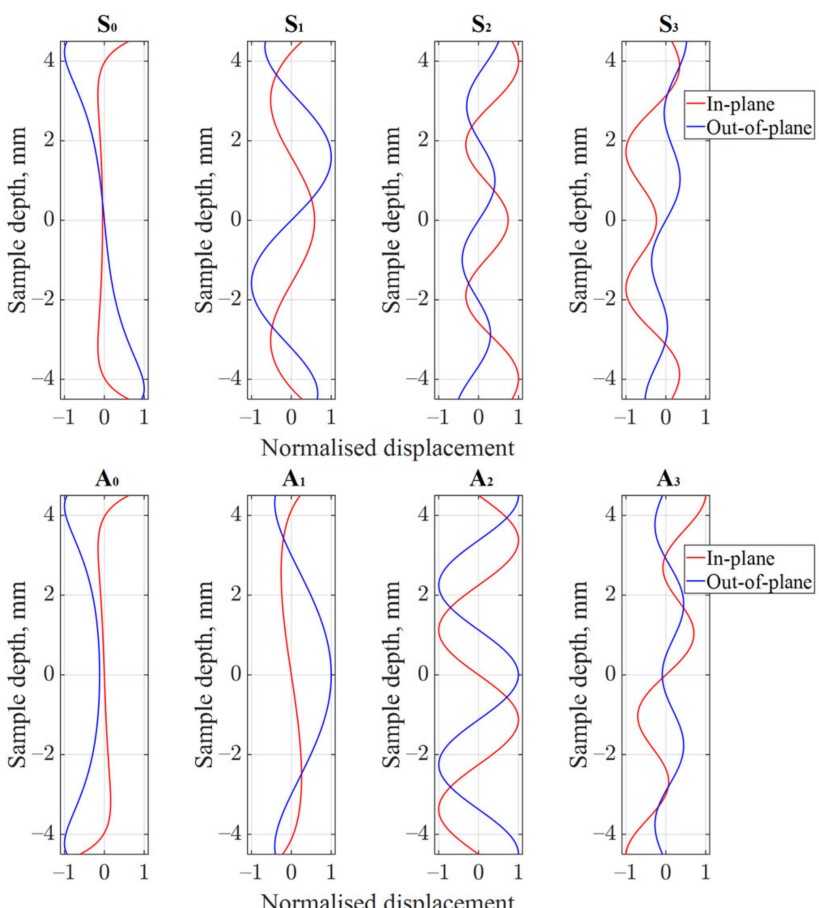

**Figure 5.** Normalized displacement of wave modes in a 9 mm steel plate at 1 MHz. The title of each figure denotes mode shapes, while indices correspond to mode order.

## 4. The Model for Analysis of $S_3$ Mode Excitation and Propagation

To verify the assumptions made on the previous section, 2D finite element simulations were carried out. The models aimed at investigating excitability of $S_3$ mode and capabilities to avoid generation of undesired co-existing modes while using conventional ultrasonic phased arrays. The geometry of the investigated structure was a rectangle with wave propagating in the direction of longest side along *x* axis. To avoid interfering reflections from edges of the plate, the length of the model increased to 3.5 m, leaving the monitored segment equal to 2 m as shown in Figure 6. Thus, the overall dimensions of the model were 9 mm × 3.5 m. In this case, we avoided using absorbing boundaries, as in some cases they still can produce weak reflections [41]. To confirm the correctness of the pipe approximation, FE models at particular cases will be generated for pipes as well.

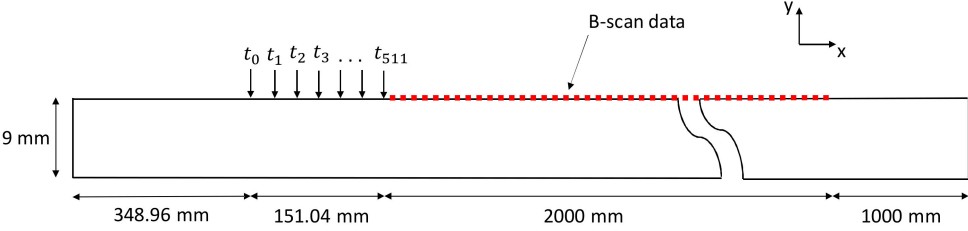

**Figure 6.** FE model setup of unrolled pipe. $t_0 - t_{511}$ indicates different excitation delays provided to nodes of mesh.

2D linear plane strain finite element model was implemented using Abaqus explicit solver (Dassault Systèmes, Vélizy-Villacoublay, France), assuming that the object is infinite in the *z* direction. CPE4R four node plane strain elements with reduced integration were chosen for the regular mesh model. The edge length of an element was set to 0.295 mm, which is λ/10 of the slowest mode in the considered frequency band. The integration step in the time domain was set to 0.03 µs, which is 1/33 of the period at 1 MHz central frequency. Both mesh size and integration step met the accuracy requirements of FE structural mechanics problems [36]. The propagating wave in *x* direction was excited by applying normal to surface force to specific number of nodes on the surface of plate starting at 32 nodes and ending with 512 nodes. This approach has been selected in order to facilitate the phased array excitation with varying aperture as it was shown that the size of array active area determines the bandwidth of excited modes along the phase velocity axis [36]. The example of 1 MHz 10 period Gaussian tone-burst excitation signal that was used throughout the simulations is presented in Figure 7. The variable monitored in this study was an in-plane (*x*) and out-of-plane (*y*) component of particle velocity at element nodes located on top surface at distances from 500 mm to 2500 mm from the origin as shown in Figure 6.

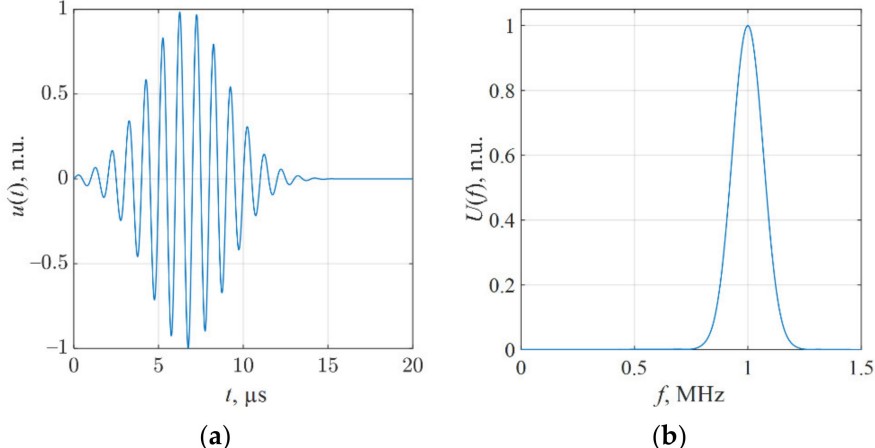

**Figure 7.** (**a**) Excitation signal waveform; (**b**) excitation signal spectrum.

*Selective Excitation of $S_3$ Mode*

As the guided waves possess many modes in the particular frequency bandwidth, the selective generation should be adjusted both in frequency and phased velocity domain. The frequency bandwidth is determined by the characteristics of ultrasonic transducer and parameters of excitation signal. More complicated is the optimization according to the phase velocity or, in other words, the spatial frequency domain. Unidirectional selective guided wave excitation minimizing other modes at the same time can be implemented by simulating the performance of the corresponding angled probe. As it was shown by other authors, angled plane wave front can be obtained with a phased array where the angle of the wavefront or delay time of array channels determines the specific phased velocity of the wave. Delay times (focal laws) for excitation of mode with particular phase velocity can be calculated using the following equation [35,42]:

$$t_i = p \cdot \frac{(i-1)}{c_{\text{ph}}}, \tag{1}$$

where $t_i$ is the delay for each element, $i$ is the array element number, $p$ is the pitch, and $c_{\text{ph}}$ is the targeted phase velocity.

While the spectrum of the excitation signal determines the limits of the frequency band in which the guided waves are generated, the aperture of the transducer determines the ranges along the phase velocity axis.

In the case of rectangular harmonic burst, the frequency bandwidth of the main lobe of the spectrum can be estimated as $\frac{2}{t_B}$, where $t_B$ is the duration of the burst. Therefore, in order to have a frequency bandwidth of around 0.2 MHz, the duration of the burst should be 10 µs. In case of a 1 MHz excitation frequency, it will correspond to 10 periods burst. In a similar way, it can be shown that the phase velocity bandwidth of generated modes can be determined according to:

$$\Delta c \sim \frac{2 \cdot c_0^2}{L_{ex} \cdot f_0},\tag{2}$$

where $L_{ex}$ is the length of excitation zone, $f_0$ is the central frequency, and $c_0$ is the velocity at the central frequency. The phase velocity of $S_3$ mode at 1 MHz is 6250 m/s according to the estimations presented in previous section. Then, the phase velocity bandwidth $\Delta c$ for this mode, estimated according to Equation (2), will look like the one presented in Figure 8.

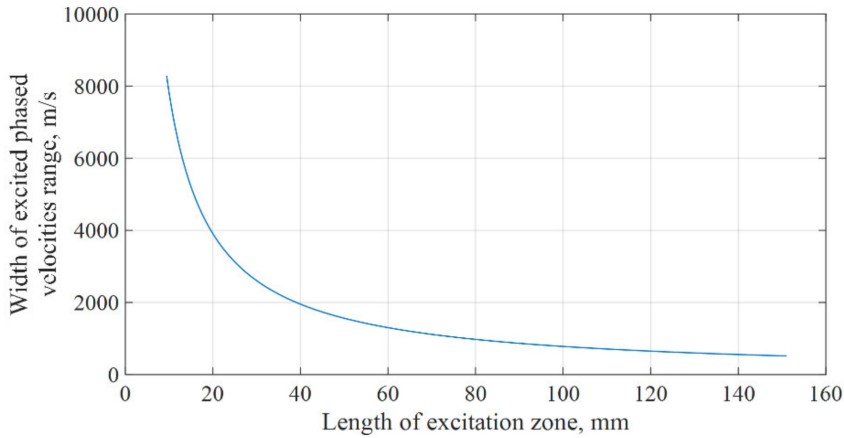

**Figure 8.** Bandwidth of excited phase velocities versus the length of excitation zone.

In order to verify this assumption, excitation of $S_3$ mode was investigated using fixed delay law, corresponding to phase velocity 6250 m/s, but at the same time, changing the length of excitation zone from 9.44 mm to 151.04 mm. In such case, the phase velocity bandwidth varies from 8300 m/s to 500 m/s, which is expected to limit number of generated modes. Meanwhile the frequency bandwidth remained the same in all cases—0.2 MHz at −6 dB level. In each of the models, the B-scan data of the out-of-plane component on the surface of plate along the $x$ direction was collected. To illustrate the propagating modes, 2D Fourier transform was used to transfer data from the time–distance to the phase velocity–frequency domain [43,44]. The obtained results are presented in Figure 9. It can be observed that in case of small apertures (9.44 mm and 18.88 mm) all modes are excited in the frequency band of input signal, showing wide phase velocity ranges in Figure 9b,d. The phase velocity range is effectively reduced using a 75.52 mm excitation zone Figure 9h. Further increase of the aperture provides even better mode isolation. In reality, the aperture of the array cannot be unlimited, meaning that other undesired modes are likely to be excited, however, with essentially smaller amplitudes. For example, even in the case of a 151 mm aperture, patterns of $S_3$, $S_2$, $S_0$, $A_4$, $A_3$, $A_2$, and $A_0$ modes can be also observed in Figure 9i,j.

This can be explained by the fact that the rectangular distribution of amplitudes in the excitation zone will lead to the appearance of relatively high side lobes in the spatial spectrum. The amplitude of these side lobes can be reduced by smoothing the excitation amplitudes at the edges of excitation zone using apodization. An apodization function (also called a tapering function or window function) is a function used to smoothly bring the amplitudes of the signal down to zero at the edges of the excitation zone. The apodization law is illustrated in Figure 10. Such excitation scenario was implemented for the case of 151 mm excitation zone only to elucidate the improvement in suppression of co-existing

modes. The results presented in Figure 11a,b show that a very weak pattern of $A_3$ and $S_2$ modes still can be observed, while other modes like $S_0$, $A_4$, $A_2$, and $A_0$ vanished. In order to demonstrate the validity of the unrolled pipe approximation, the same excitation with apodization was implemented for a 2D pipe structure with circumference of 2 m. The results presented in Figure 11c,d found to be similar to the ones obtained on a plate. Similar patterns of $A_3$ and $S_2$ modes as well as wave crossing points can be observed, suggesting the validity of the plate approximation.

In order to suppress the existing $A_3$ asymmetric mode, excitation from both sides of the plate with the same apodization was implemented instead of increasing excitation zone. The results of simulation can be seen in Figure 12. It can be observed, that in this case, a pure $S_3$ mode is propagating in the structure. While such two-sided excitation approach is unpractical, it has big advantage in the modelling as it enables to analyze separately interaction of $S_3$ mode with defects. In this way, only $S_3$ mode can be explored avoiding the presence of other modes that may lead to contradicting results.

Taken together, these results demonstrate, that suppression of co-existing modes can be achieved by implementing delays to subsequent nodes, apodization, and two-sided excitation. In the next section, the excitation of $S_3$ and co-existing modes are explored with experiments.

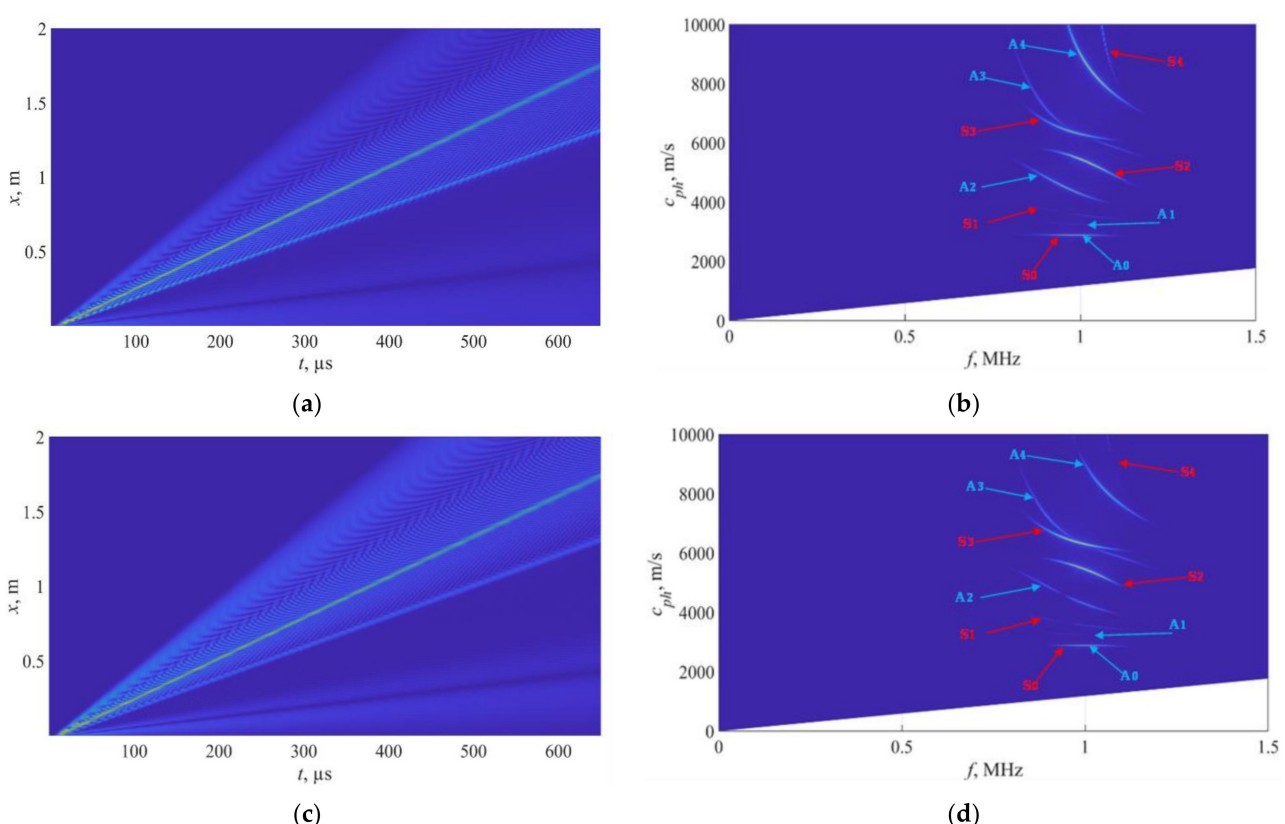

**Figure 9.** *Cont.*

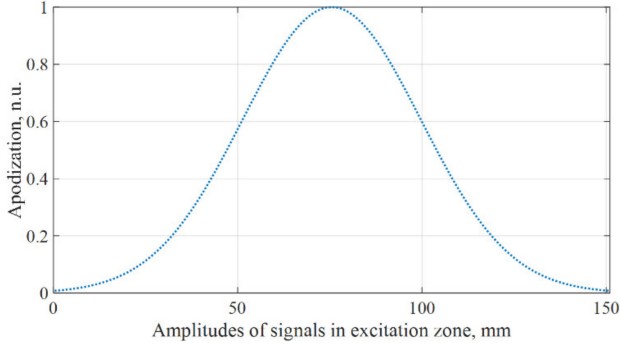

**Figure 9.** B-scans and reconstructed phase velocities in case of different length excitation zone: (**a**,**b**) 9.44 mm; (**c**,**d**) 18.88 mm; (**e**,**f**) 37.76 mm; (**g**,**h**) 75.52 mm; (**i**,**j**) 151.04 mm.

**Figure 10.** Apodization law versus excitation zone.

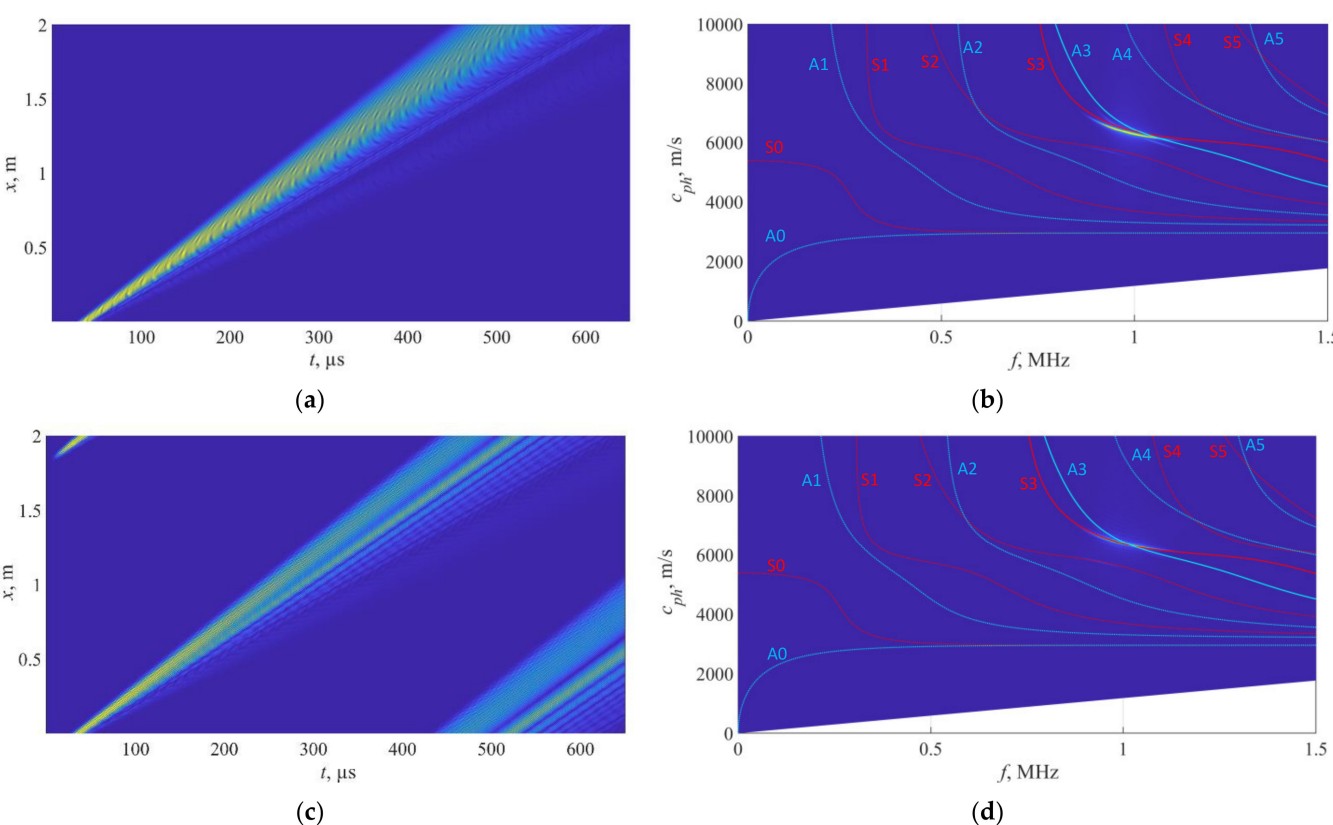

**Figure 11.** Signals in the case of excitation with apodization: B-Scan of the vertical component of the particle velocity simulated on a (**a**) plate and (**c**) corresponding pipe; phase velocity dispersion curves reconstructed on (**b**) plate and (**d**) pipe.

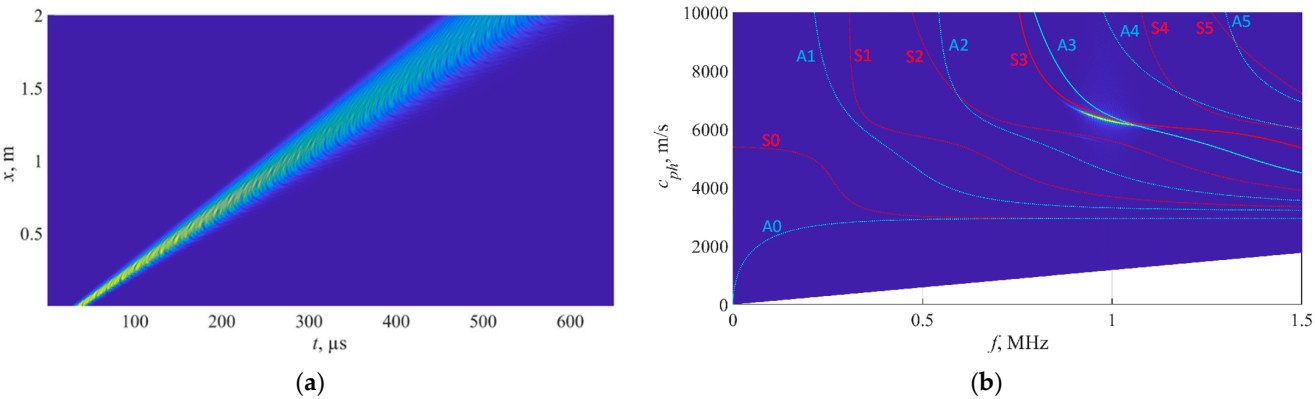

**Figure 12.** Signals in case of excitation with apodization from both sides: (**a**) B-Scan from top side of the plate; (**b**) reconstructed phase velocity dispersion curves.

## 5. Experimental Validation

To verify the simulation results, which demonstrated possibility to excite symmetric $S_3$ mode while suppressing co-existing modes, experiments were carried out. For this purpose, 10 mm thickness SteelAlloy1020 plate with dimensions 1000 mm × 2000 mm was selected. Two 1 MHz 32 element Imasonic CdC9463-2 phased arrays with pitch of 2.05 mm and active aperture of 65.1 mm were arranged in pitch-catch configuration and placed 1000 mm apart. The bandwidth of the selected probe is 57% at –6 dB. To generate and record signals of propagating modes, the multichannel data acquisition system Dasel Sitau (Dasel Sistemas, Madrid, Spain) with 128 parallel channels was used. The view and schematic diagram of experimental arrangement are presented in Figures 13 and 14, respectively.

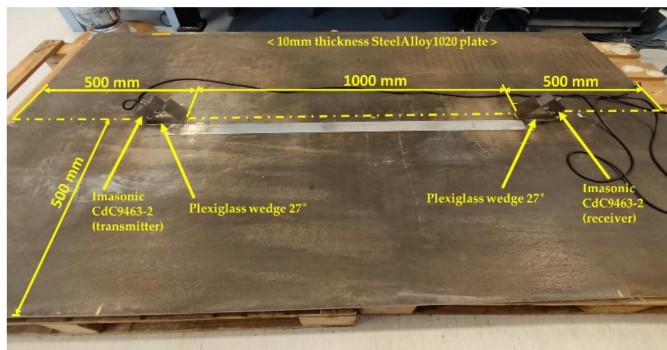

**Figure 13.** Photo of the experimental S$_3$ guided wave excitation in a 10 mm steel plate.

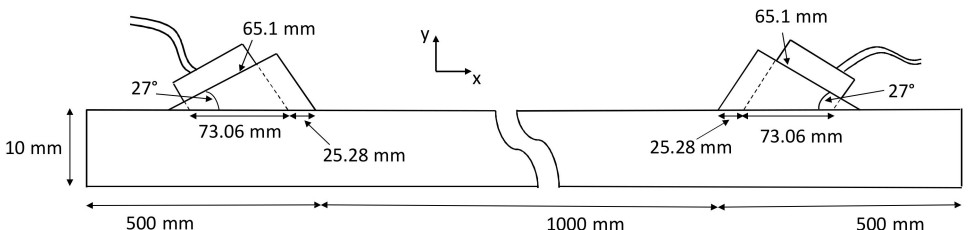

**Figure 14.** Experiment setup of 10 mm steel plate, selective higher mode excitation with angled probe with phased array.

Throughout the experiments, special 27° plexiglass wedges were manufactured and used as the interface between the array and the sample. Such specific wedge angle was selected according to the excitation angle curves, which were calculated for 10 mm thickness steel plate and are presented in Figure 15. According to the results, presented in Figure 15, the excitation angle for S$_3$ mode at 1 MHz is 26.35° where phase velocity is equal to 6127 m/s. In order to set proper incidence angle, the wavefront of the plane wave generated by phased array was slightly steered by 0.65° changing delay laws $t_{st_i}$ of each array element in order to achieve exactly 26.35° incidence:

$$t_{st_i} = p \cdot \frac{(i-1)\sin\alpha_{st}}{C_L}, \tag{3}$$

where $t_{st_i}$ is the steering delay of each element, $p$ is the element pitch, $i$ is the element number, $C_L$ is the longitudinal velocity in plexiglass prism, $\alpha_{st}$ is the steering angle. At 27° angle, a projection of the wave-front on the surface of the steel plate is equal to 73.06 mm, and this is considered as array aperture throughout this experiment.

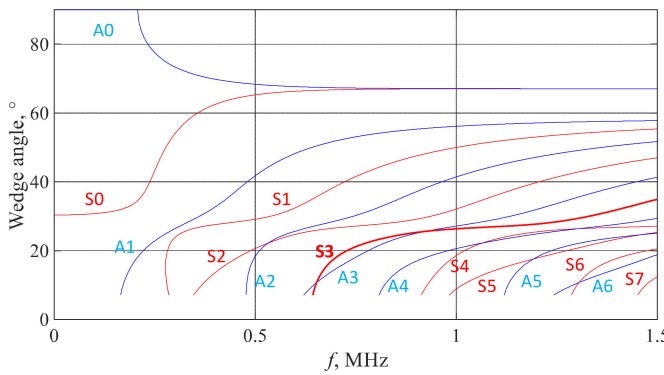

**Figure 15.** Excitation angle of the specific wave mode using plexiglass wedge.

Since the experiment was carried out on a 10 mm plate while the simulations presented earlier were carried out on 9 mm thickness sample, a new FE model was conducted here which mimics the conditions of the experiment. In this model, a zone equal to 73 mm was excited using normal to surface force, and particle velocity at the nodal points located 1123.5 mm away from the transmission zone were collected. Such distance takes into account the dimensions of the Plexiglas wedges and 1000 mm distance between them corresponding to the experiment. The schematic diagram of FE model is presented in Figure 16.

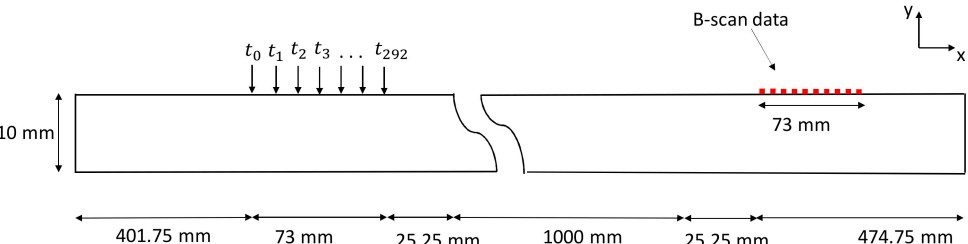

**Figure 16.** FE model setup for matching with experimental one.

During experiments, the signals were recorded separately by each element. Both, experimental and modelled signals then were integrated, considering the phasing angle and in such a way simulating performance of the phased array. Integration was implemented according to the following equation:

$$u_{\Sigma}(t) = \sum_{i=1}^{K} u_i(t - t_{\text{st},i}), \tag{4}$$

where $u_i(t)$ is the signals measured by $i$th phased array element; $K$ is the total number of elements in phased array; $t_{st,i}$ is the phasing delay calculated according to Equation (3). The signals obtained from the FE model and experiment are presented in Figure 17.

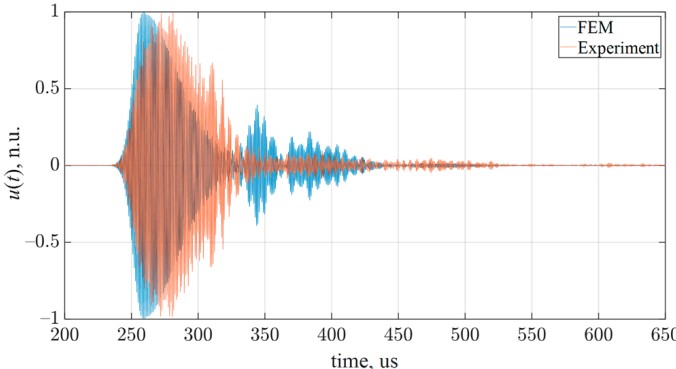

**Figure 17.** Comparison of simulated and experimental $S_3$ mode signals.

Simulation and experimental results show good agreement for $S_3$ wave mode. A significant difference can be observed at the trail of the signals, which is due to the appearance of slower modes. From the 2D FFT data of FE model (Figure 18), it can be observed that weak patters of $S_2$, $A_3$, $A_4$, and $S_4$ modes are still visible. However, as the $S_3$ mode exhibits high group velocity, it will not overlap with co-existing modes at sufficient distances. The generation of parasitic $A_4$, $S_4$, and $S_2$ modes can be suppressed by limiting the frequency bandwidth of the probe while increasing the cycle count of the excitation signal. $A_4$, $S_4$, and $S_2$ mode generation is also determined by limited size of the active aperture.

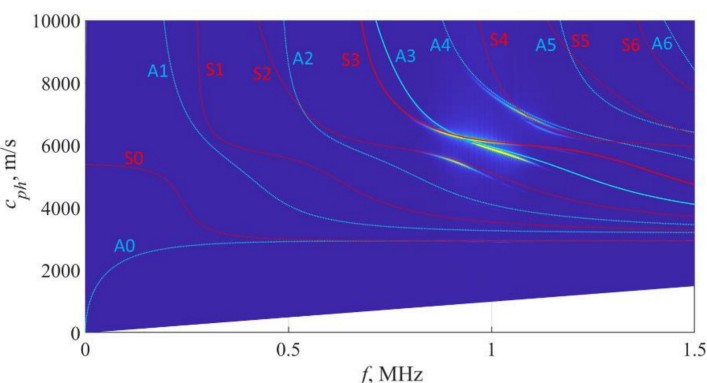

**Figure 18.** Reconstructed phase velocity dispersion curves from FE results on 10 mm plate.

## 6. Interaction of $S_3$ Mode with Corrosion Equivalent Defects

Having outlined the conceptual principles of excitation of isolated $S_3$ mode, in this chapter, mode response to synthetic corrosion-like defects is investigated. 2D FE models with different extent of wall thickness loss were used for a virtual experiment. While the unrolled pipe approach has been validated earlier, synthetic models represented a plate scenario. In total, three different types of defects that mimic wall thickness loss due to corrosion were considered, namely, localized, pitting, and uniform. Localized corrosion was simulated as cylindrical cut with radius of 50 mm and depth of 30% and 50% of the initial wall thickness of the model. Pitting defect was considered as a cluster of 3 mm wide rectangular cuts evenly spaced at 50 mm zone. The pitch between the cuts was set to 10 mm, leading to 6 successive defects in the considered area. Two cases of pitting defect were executed, 30% and 50% of the total wall thickness. Finally, the uniform defect was implemented as 1000 mm zone with gradual wall thickness loss, reaching a 50% of the initial wall thickness at 500 mm distance. Cases considered at simulations are illustrated in Figure 19.

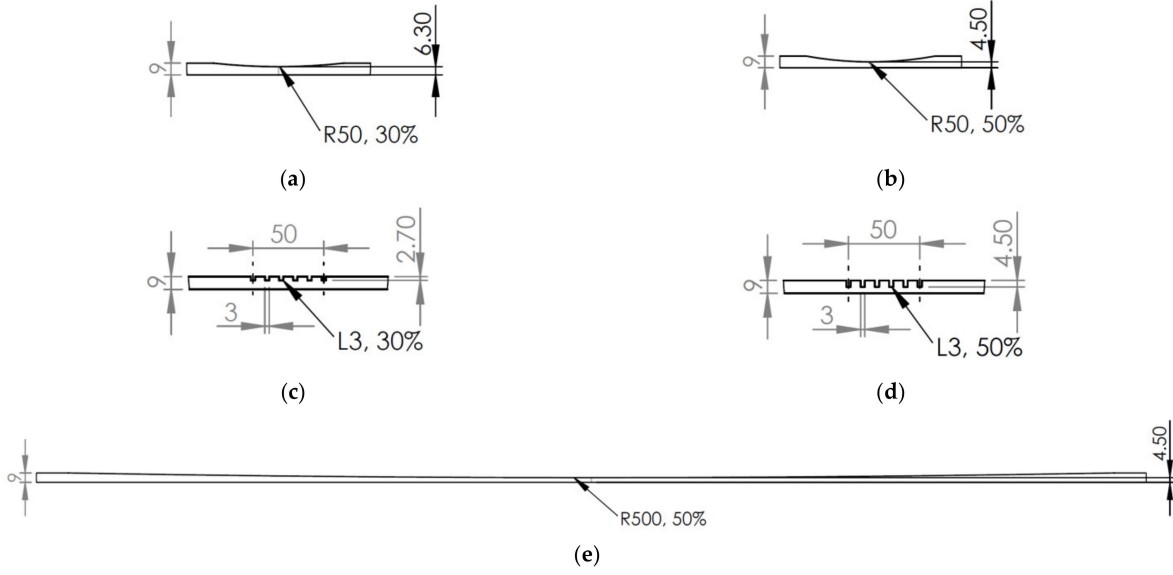

**Figure 19.** Defect geometries considered in virtual experiment: (**a**,**b**) localized defect with 30% and 50% wall thickness loss, (**c**,**d**) pitting defect with 30% and 50% of wall thickness loss, and (**e**) smooth wall thickness loss at 1000 mm zone with maximum depth of 50% of the wall thickness.

The calculations were executed using Abaqus explicit FE solver (Dassault Systèmes, Vélizy-Villacoublay, France), creating a 2D plane strain FE model for each case. The propagation of the $S_3$ mode was simulated in the 2000 mm zone, while the defects were centered

at 1000 mm. The uniform mesh size of 0.295 mm was used in the areas outside the defects. In the defect regions, the edges of the model were seeded providing a minimum allowed element size of 0.16 mm. The integration step in the time domain was set to 0.03 µs, which is 1/33 of the period at 1 MHz central frequency. The wave propagating in *x* direction was excited by applying normal to surface force to a 151.04 mm zone using the symmetrical two-sided excitation and apodization as it was discussed in chapter 4. The $S_3$ mode was excited with a 10 period 1 MHz Gaussian tone-burst excitation signal. The variable monitored in this study was out-of-plane (*y*) component of particle velocity at nodes located on the outer surface of the plate. Figure 20 shows the obtained Bscans of the out-of-plane component and corresponding 2D FFT images in case of different types of defects.

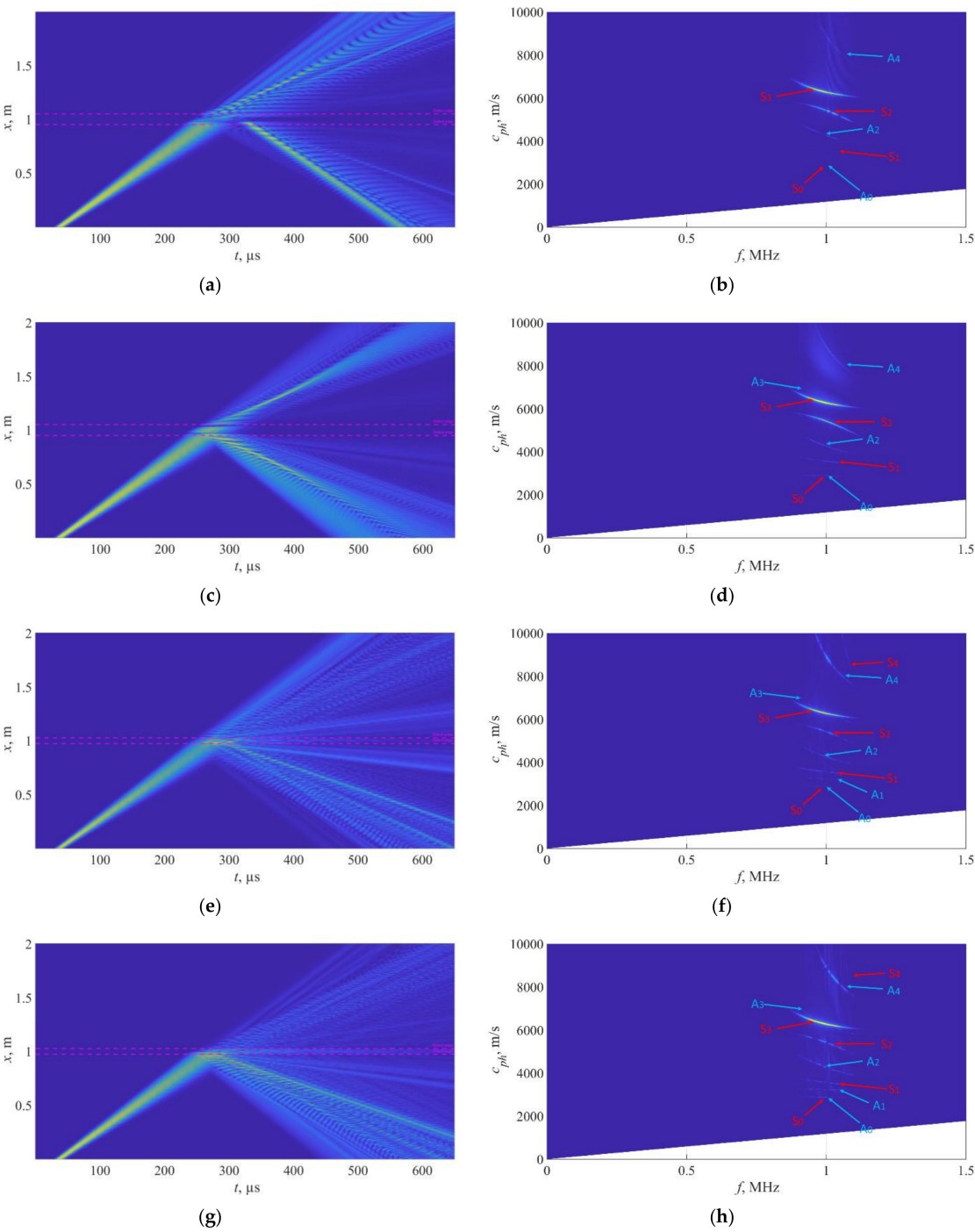

**Figure 20.** *Cont.*

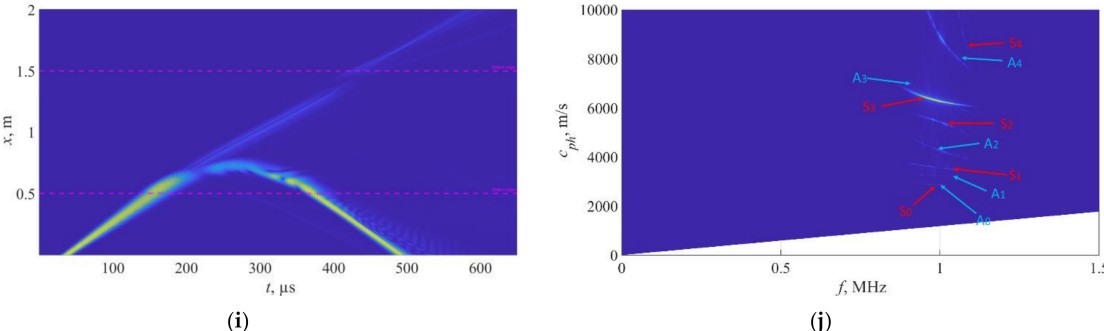

**Figure 20.** The B-scans and reconstructed phase velocities in case of corrosion-like defects: (**a**,**b**) localized defect of 30% wall thickness; (**c**,**d**) localized defect of 50% wall thickness; (**e**,**f**) pitting defect of 30% wall thickness; (**g**,**h**) pitting defect of 50% wall thickness; (**i**,**j**) uniform defect of 50% wall thickness.

Various defects show different outcomes of the interaction with $S_3$ mode. For example, localized defects (Figure 20a–d) produce quite strong converted $S_2$ mode, which propagates with lower group velocity and arrives after the direct $S_3$ mode. The presence of $S_2$ mode in case of pitting defects (Figure 20e–h) is no longer that evident. It can be speculated that multiple scattering and mode conversion takes place at each cut of pitting defect providing a diffused ultrasonic signal at the output. A uniform defect (Figure 20i,j) does not show any mode conversion; however, significant delay of the $S_3$ arrival can be expected. The presented results suggest the usefulness of the $S_3$ mode in detecting defects that mimic different types of corrosion. The detailed analysis of the $S_3$ mode response to defects was out of the scope of this paper and remains a subject for the follow-up paper.

## 7. Conclusions

In this paper, higher order symmetrical modes suitable for corrosion detection were analyzed, employing the dispersion relations, in-plane and out-of-plane distributions, and leakage to surrounding media. It was determined, that $S_3$ mode could be a good candidate for corrosion assessment, as it has sufficiently short wavelength, possess relatively low attenuation and can be excited and received using conventional ultrasonic phased arrays. Finite element simulations demonstrated that it is possible to generate a pure $S_3$ mode, by using phased excitation with a sufficiently large aperture, by decreasing side lobes using apodization towards the edges of excitation zone, and by implementation of two-sided excitation to avoid the presence of asymmetric modes. The model obtained in this study can be used for further analysis of the $S_3$ mode interaction with defects typical to corroded structures.

It was demonstrated through experiments that the $S_3$ mode can be generated in the structure using conventional phased arrays. A good agreement was found between the modelling and the experiments. Despite the experiments demonstrated presence of parasitic modes, the high group velocity of $S_3$ allows to distinguish it from the surrounding signals quite easily at sufficient propagation distances. The findings presented in this paper suggest using a high order symmetric mode for corrosion assessment and demonstrate how different techniques based on phased array or angled excitation could be used in order to excite a selected wave mode into the structure minimizing generation of undesired waves.

Finally, it has been demonstrated that $S_3$ mode is sensitive both to localized and distributed defects that correspond to the corrosion of metallic structures. To date, this mode has been used for such defect detection for the first time and can be used for rapid screening around the circumference of the pipe. The detailed interaction between the $S_3$ mode and defects of various types remains the subject of a follow-up paper.

**Author Contributions:** Conceptualization, D.C.; methodology, V.S., L.M., R.R., and E.Ž.; software, V.S., and L.M.; validation, D.C., and L.M.; formal analysis, D.C.; investigation, D.C., V.S., and E.Ž.; resources, D.C., V.S., L.M; data curation, R.R.; writing—original draft preparation, D.C.; writing—review and editing, D.C., V.S., L.M., R.R., and E.Ž.; visualization, D.C., V.S.; supervision, V.S. All authors have read and agreed to the published version of the manuscript.

**Funding:** This research was funded by the Research Foundation of the Research Council of Lithuania under the project POLITE "Assessment of the distributed and concentrated pipeline corrosion by means of ultrasonic testing and machine learning methods", No. MIP2113.

**Institutional Review Board Statement:** Not applicable.

**Informed Consent Statement:** Not applicable.

**Data Availability Statement:** Not applicable.

**Conflicts of Interest:** The authors declare no conflict of interest.

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
