# Peer review of "Selection of Higher Order Lamb Wave Mode for Assessment of Pipeline Corrosion"

_metals, doi:10.3390/met12030503_

Round 1

Reviewer 1 Report

The objective of this work was to investigate higher order modes for corrosion detection and to determine the most promising ones in sense of excitability, leakage losses,propagation distance and potential simplicity in the analysis. In my opinion, the article is clearly written. The authors describe in detail and with interesting scientific details the experiment performed. The introduction is sufficient and the research aim is clear, the methodology is suitable, the results are presented in a proper manner, their discussion is more or less sufficient, the conclusions, in part, represent the main achievements. The article is well structured and the mathematical apparatus presented really supports the theoretical part necessary to describe the experiments. 
The figures are made suggestively, and the tables presented contain important data for the study presented. I honestly have no corrections to make to improve this material. I admit that this field has been and is intensely studied, it does not present a novelty in research, but the way of presenting and the dedication with which this material seems to have been worked deserves my acceptance for publication.  
I recommend the author to check one more time the bibliography and to change somehow figure no. 13. As this has no scientific relevance in my opinion being in fact an experimental stage that I do not doubt.
Best regards, 

Author Response

Response of the authors

Ref: metals-1611192,                                                       

Title: Selection of higher order Lamb wave mode for assessment of pipeline corrosion

Dear Reviewer,

Following your letter regarding the manuscript ref.: metals-1611192 „Selection of higher order Lamb wave mode for assessment of pipeline corrosion” submitted to Metals, we are sending the rebuttal letter explaining the changes we have performed in the text of the manuscript. We have studied carefully the review and would like to thank for valuable remarks and comments. We agree with all remarks and proposals. Comments to remarks are listed below:

Revisions in the manuscript concerning the Reviewer #1 comments are highlighted in green.

  1. I recommend the author to check one more time the bibliography and to change somehow figure no. 13. As this has no scientific relevance in my opinion being in fact an experimental stage that I do not doubt.

Response of the authors: We agree with this thoughtful remark. We have checked the bibliography one more time. We updated Figure 13 (line 426) to make it more informative for the readers.

Once again we would like to thank the reviewer for insightful comments. We hope that the revisions in the manuscript and our accompanying responses will be sufficient to make our manuscript suitable for publication in Metals.

Reviewer 2 Report

It is a good-quality paper. It addresses an important problem and presents interesting numerical and experimental studies.

The main critical point is regarding the theoretical framework, which utilises dispersion curves for plates, which is not appropriate if one side of the structure is exposed to liquid. The different boundary conditions lead to generation of different type of guided waves instead of Lamb waves, e.g. quasi-Scholte (QS) waves. The reviewer strongly recommends to study the latest publications [1-3], which describe the correct theoretical framework for the problem under consideration and made some adjustments to the paper.

1. Numerical and experimental investigations on mode conversion of guided waves in partially immersed plates; Measurement, Volume 190, 28 February 2022, 110750; doi.org/10.1016/j.measurement.2022.110750

2. Ultrasonic Guided Wave Field Modeling in a One-Side Water-Immersed Steel Plate; EASEC16, 1131-1140; DOI:10.1007/978-981-15-8079-6_106

3. Scattering characteristics of quasi-Scholte waves at blind holes in metallic plates with one side exposed to water;  NDT & E International 117, 102379, doi.org/10.1016/j.ndteint.2020.102379

Author Response

Response of the authors

Ref: metals-1611192,

Title: Selection of higher order Lamb wave mode for assessment of pipeline corrosion

Dear Reviewer,

Following your letter regarding the manuscript ref.: metals-1611192 „Selection of higher order Lamb wave mode for assessment of pipeline corrosion” submitted to Metals, we are sending the rebuttal letter explaining the changes we have performed in the text of the manuscript. We have studied carefully the review and would like to thank for valuable remarks and comments. We agree with all remarks and proposals. Comments to remarks are listed below:

Revisions in the manuscript concerning the Reviewer #2 comments are highlighted in yellow.

  1. The main critical point is regarding the theoretical framework, which utilises dispersion curves for plates, which is not appropriate if one side of the structure is exposed to liquid. The different boundary conditions lead to generation of different type of guided waves instead of Lamb waves, e.g. quasi-Scholte (QS) waves. The reviewer strongly recommends to study the latest publications [1-3], which describe the correct theoretical framework for the problem under consideration and made some adjustments to the paper.

Response of the authors: We agree with the reviewer on this important point. We have studied the provided and other latest publications related to the topic. An additional paragraph has been added to the text of the article (lines 169-180), addressing the issue of liquid load. Additionally, two new references have been added (lines 613-617).

We agree, that liquid loading leads to generation of other waves like quasi-Scholte or leaky Rayleigh wave. However, at our frequency range of interest (around 1MHz for 9mm plate), the quasi-Scholte wave is in non-dispersive region, thus the in-plane displacements in fluid are dominant and the wave can no longer be detected from surface of the plate. Moreover, since it has lower group velocity compared to high order symmetrical modes, we believe that quasi-Scholte wave in any way will not interfere or affect the propagation of the considered S3 mode. However, we agree with the reviewer that the article should demonstrate our overall understanding of traction-free and liquid loaded cases, hence we added appropriate references and paragraph to the article.

Once again we would like to thank the reviewer for insightful comments. We hope that the revisions in the manuscript and our accompanying responses will be sufficient to make our manuscript suitable for publication in Metals.

Round 2

Reviewer 2 Report

The manuscript can be accepted for publication in the present form. Minor language corrections will improve the presentation.

This manuscript is a resubmission of an earlier submission. The following is a list of the peer review reports and author responses from that submission.

Round 1

Reviewer 1 Report

This paper has investigate higher order modes for corrosion detection, employing the dispersion relations, in-plane and out-of-plane distributions and leakage. Due to the difference in curvature between the inner and outer surfaces, the high-order circumferential mode in the pipe is very different from the high-order lamb mode in the plate . It is not appropriate to simplify the pipe circumferential wave model to a plate model directly in this paper. The topic of the paper is relevant to the ultrasonics guided waves, while I did not see any serious errors in the results, there was insufficient novelty here to warrant a journal paper:

  • Line 47: What do you mean with:”This theoretically that allows to detect defects starting at 3-3.15mm.”. The detection sensitivity of bulk waves is one-half of the wavelength, but the sensitivity of guided waves is not. Please explain.
  • “To simplify the SAFE model, assumption has ….” In line152,this paper simplifies the pipe model directly to a flat plate model without giving a specific explanation. As well known, the mode crossing phenomenon (It can be observed from Fig 11 and 12) for higher order modes in the plate, where the symmetric mode and the adjacent antisymmetric mode intersect. However, this mode crossing phenomenon vanishes in the circumference waves of the hollow cylinder. Therefore, it is not appropriate to directly use the high-order modes in the plate to approximate the high-order circumferential wave in the pipe.
  • Line 199:”This means that either thickness mode transducers or phased arrays can excite and receive modes that has sufficient out-of-plane component only.”, this needs more explanation. Also what is a thickness mode transducers?
  • Line 218-219:” The leakage losses of S3 mode at 1MHz is approximately 34dB/m, which makes it appropriate for inspection at distances up to 1.5-2m.”, Please explain.
  • Line 309-310: Eq.(2) give the phase velocity bandwidth of generated modes.To facilitate the reader's understanding, the detail of derive in manuscript should be illustrated.
  • Line 280:”the phased array excitation with varying aperture” Please add the diagram to illustrate the aperture.

Reviewer 2 Report

1.  In the paper, the author should describe the test instruments used in the experiment.

2. What is the measurement accuracy of the method proposed in this paper? How much is the measurement error compared with the actual situation? It is best to compare with other methods.

3.What is the basis for setting the experimental method in Figure 16.

4.The references cited in this paper can not reflect the innovation of this method. 

Reviewer 3 Report

The authors have presented a scenario of using S3 Lamb wave mode for inspection of pipes. While the feasibility study on S3 mode with FEM is interesting, it seems that the presented work is unfinished. The design of the experiment to validate the usage of the S3 Lamb wave mode is not complete. For example, referring to leakage losses, the authors have not reported any FEM or experimental study of the pipe to be in contact with liquids. This could have been compared by using other modes so that readers would see the real difference by results and not only by calculations. Meanwhile, the authors talk about the assessment of pipeline corrosion, while they have not conducted experiments regarding any corrosion detection. In addition, a good experiment and FEM study should have been done on a pipe and not on a plate. My comments for the rest of the manuscript are as follow:

  1. Although maybe well-known to researchers in the ultrasonic and NDT field, the authors are recommended to define abbreviations such as NDT, EMAT, FEM, SAFE and SH in the introduction.
  2. In the introduction, line 49, please add the missing reference for micron levels and repeatability.
  3. In the introduction, line 60, “In case of small localized defects or smooth wall thinning, the response produced by the flaws are usually barely detectable by currently existing guided wave technology [16].”, the reference [16] does not discuss guided waves. Are the authors sure that is relevant? Please check.
  4. In the introduction, line 84, “ The major downside of the proposed set-up is requirement of EMAT transducer in order to generate SH wave.” Please explain why the requirement of EMAT is a downside.
  5. In the introduction, line 107, “Symmetrical modes usually have less leakage losses”, please add missing references.
  6. Please summarise the material properties and dimensions of the sample used in the experiment in a table.
  7. In section 3, line 155, please explain based on what criteria 20 discrete elements with the size of 0.45 mm was chosen. Please also elaborate on the element type composed of 3 nodes.
  8. In figure 1, please add a label for each curve (mode) so that readers will see which curve is S1, S2 and so on. As there are symmetrical and six asymmetrical modes, it would be better to scale up the figure so that the labels are not confusing. The same could be done for figure 2 and figure 3.
  9. In line 167, “However, it’s known that at such frequencies the out of plane displacements of particular modes are usually poor, making them difficult to excite and receive using conventional ultrasonic transducers” please add the missing reference. Please also avoid using contractions such as “it’s”.
  10. In line 175, “On the other hand, the modes which possess strong out-of-plane component will be strongly attenuated due to leakage losses as most of the pipelines are filled with various substances”, please add the missing references.
  11. In line 183, “Additionally, the phase velocity of S3 mode at 1MHz is equal to 6000-6300m/s which results in wavelength varying from 6mm to 6.3mm.” how this wavelength is expected to generate guided waves? Is having a greater wavelength than the thickness of the structure, not a condition of generating guided waves? Please explain concerning the following work:
  • Rose, Joseph L. Ultrasonic guided waves in solid media. Cambridge university press, 2014.
  1. In line 185, “This theoretically that allows to detect defects starting at 3-3.15mm.” is the value based on half- wavelength? Please explain.
  2. In line 197, please add an appropriate citation for your statement.
  3. What is the unit for the y-axis in figures 2, 10 and 17?
  4. In line 255, “The results presented above suggest that S3 mode at 1MHz could be used for corrosion assessment as it has high group velocity, sufficient out-of-plane displacements and acceptable leakage losses for inspections up to 2m” please quantitively define “sufficient” and “acceptable” as they are subjective words.
  5. In line 266, “To avoid interfering reflections from edges of the plate the length of the model increased to 3.5m”, please explain why instead of using a damping boundary condition on edges, the length of the model was increased?
  6. Please summarise the FEM parameters in a table.
  7. In line 275, where did the λ/10 rule of the slowest mode and 1/33 of the period at 1 MHz come from?
  8. How can the authors be certain that applying a “normal to surface force to a specific number of nodes” will give the same results as having a Multiphysics simulation of transducers?
  9. Is the statement in line 305 and equation 2 achieved by the authors or taken from elsewhere? Please check if you need to add citations.
  10. In the experimental validation section, line 363, the selection of “10 mm thickness” is different from “9mm thickness” in line 134 and 9 mm thickness in FEM. If the experimental sample has 10 mm thickness, why did not the authors conduct a 9 mm thickness FEM in the first place? FEM in section 5 could be moned to section 4 so that FEM studies have a separate section.
  11. How was figure 15 generated?
  12. Can the authors add a quantitative analysis on the difference of signals in FEM and the experiment in figure 17? Please also consider not overlapping two signals in figure 10.